# Nanostructured Polymethylsiloxane/Fumed Silica Blends

**DOI:** 10.3390/ma12152409

**Published:** 2019-07-28

**Authors:** Iryna Protsak, Volodymyr M. Gun’ko, Volodymyr V. Turov, Tetyana V. Krupska, Eugeniy M. Pakhlov, Dong Zhang, Wen Dong, Zichun Le

**Affiliations:** 1College of Environment, Zhejiang University of Technology, Hangzhou 310014, China; 2College of Science, Zhejiang University of Technology, Hangzhou 310023, China; 3Chuiko Institute of Surface Chemistry of National Academy of Sciences of Ukraine, Kyiv 03164, Ukraine; 4Department of Chemical & Biomolecular Engineering, University of Akron, Akron, OH 44325, USA

**Keywords:** polymethylsiloxane/nanosilica blends, hydration effect, mechanical loading effect, textural characteristics, interfacial layer structure

## Abstract

Polymethylsiloxane (PMS) and fumed silica, alone and in a blended form (1:1 w/w), differently pretreated, hydrated, and treated again, were studied using TEM and SEM, nitrogen adsorption–desorption, ^1^H MAS and ^29^Si CP/MAS NMR spectroscopy, infrared spectroscopy, and methods of quantum chemistry. Analysis of the effects of adding water (0–0.5 g of water per gram of solids) to the blends while they are undergoing different mechanical treatment (stirring with weak (~1–2 kg/cm^2^) and strong (~20 kg/cm^2^) loading) show that both dry and wetted PMS (as a soft material) can be grafted onto a silica surface, even with weak mechanical loading, and enhanced mechanical loading leads to enhanced homogenization of the blends. The main evidence of this effect is strong nonadditive changes in the textural characteristics, which are 2–3 times smaller than additive those expected. All PMS/nanosilica blends, demonstrating a good distribution of nanosilica nanoparticles and their small aggregates in the polymer matrix (according to TEM and SEM images), are rather meso/microporous, with the main pore-size distribution peaks at *R* > 10 nm in radius and average <*R*_V_> values of 18–25 nm. The contributions of nanopores (*R* < 1 nm), mesopores (1 nm < *R* < 25 nm), and macropores (25 nm < *R* < 100 nm), which are of importance for studied medical sorbents and drug carriers, depend strongly on the types of the materials and treatments, as well the amounts of water added. The developed technique (based on small additions of water and controlled mechanical loading) allows one to significantly change the morphological and textural characteristics of fumed silica (hydrocompaction), PMS (drying–wetting–drying), and PMS/A-300 blends (wetting–drying under mechanical loading), which is of importance from a practical point of view.

## 1. Introduction

Silicas can easily be modified to significantly change many important characteristics of the materials [1,2,3,4,5,6,7,8,9,10,11,12,13,14,15,16]. Various functionalized silicas (fumed nanosilica, silica gels, precipitated silicas, ordered mesoporous silicas, etc.) and siloxane-based polymers (e.g., linear polydimethylsiloxane, PDMS, 3D-cross-linked polymethylsiloxane (PMS)) are widely used in industry and medicine as well in various branches of science [1,2,3,4,5,6,7,8,9,10,11,12,13,14,15,16,17,18,19,20,21,22,23]. One of the main characteristics of these materials is their degree of hydrophobicity, which allows their interactions with various liquids, polymers, and solids to be controlled. PMS, as a 3D-cross-linked polymer with one CH_3_ group attached to each Si atom and residual silanols [21,22,23], may be considered as an intermediate between PDMS and methylated nanosilica containing residual silanols [1,2,24,25,26,27,28]. In a series of nanostructured silicas with surface functionalities, such as –O)_3_SiCH_3_, –O)_2_Si(CH_3_)_2_, and –O–Si(CH_3_)_3_, the degree of hydrophobicity increases, i.e., PMS is less hydrophobic than PDMS. There is a commercial hydrogel with PMS incompletely cross-linked and containing a certain amount of residual silanols (a medicinal sorbent entitled Enterosgel, it contains 7–8 wt.% of PMS and 93–92 wt.% of water, is produced by Kreoma-Pharm, Ukraine, and it is also used as a drug carrier), which represents a gel-like soft material [29,30]. However, pure PDMS is hydrophobic (there are no silanols, and the siloxane bonds tend to be low-polar bonds) and cannot form hydrogels. Note that the Si–O–Si bonds are relatively flexible due to small changes in the energy which follow upon changes in the ∠SiOSi angle in the 140 to 180° range [1,2]. Amorphous silicas are therefore relatively stable despite certain differences in 3–8-member siloxane rings (characterized various values of ∠SiOSi) formed upon the synthesis. Clear, PMS is less flexible than PDMS, but PMS is soft because only three bonds from four ones of each Si atom can take part in cross-linking [29,30]. 

Clear, the properties and characteristic of any nanostructured composites strongly depend on the distributions (uniform – nonuniform, nanolayers – nanoclusters – nanodomains – microdomains – monolith fragments) of different components in the composites. These effects could be stronger (especially upon interaction with water [31,32,33,34,35,36,37]) for the composites with nanostructured hydrophilic and hydrophobic components [38,39]. Using specific treatments and certain (rather small) amounts of water, it is possible to prepare hydrophilic blends with hydrophobic and hydrophilic components. Subsequent treatments (e.g., drying) of these blends allow one to obtain hydrophobic composites, the hydrophilic properties of which can renew after additional treatment (e.g., wetting and stirring or grinding at greater mechanical loading) [38,39]. 

Fumed nanosilica and Enterosgel are used in medical applications not only as sorbents but also as drug carriers [17,29,30,40]. Therefore, changes in the textural characteristics of the drug carriers can strongly affect the drug release upon the medical applications of the drug delivery systems. The textural and hydrophilic/hydrophobic characteristics of the blends of nanosilica and PMS could be strongly changed and better controlled than those of individual sorbents due to their morphological, textural, and structural features. 

Fumed nanosilica and PMS are soft-powder materials because their aggregates of nanoparticles and agglomerates of aggregates may be easily rearranged under any treatment [36]. The results of these treatments depend on several factors such as the (i) hydrational, thermal, and mechanical history of the components; (ii) the weight ratio of the components; (iii) the amounts of added water; (iv) the mechanical loading; (v) the time of treatment; and (vi) the temperature [1,2,24,25,26,36,37,38,39,40,41]. In this study, two important factors, namely, a small amount of added water and the type of mechanical loading upon stirring of the PMS/nanosilica blends, are analyzed to control the characteristics of the final blends. 

## 2. Materials and Methods

### 2.1. Materials

Commercial polymethylsiloxane (PMS) hydrogel (Enterosgel, hPMS, Kreoma-Pharm, Ukraine, ~7–8 wt.% of PMS and 93–92 wt.% of water) and dry PMS (dPMS) were used as the initial materials. Note that after drying of Enterosgel at room temperature for a week, the amount of residual water bound in PMS is small (~0.7 wt.%), i.e., the material becomes rather hydrophobic. Fumed silica (nanosilica) A-300 (Pilot plant of Chuiko Institute of Surface Chemistry, Kalush, Ukraine) was mixed with water (1:5) and dried at 160 °C for several hours, resulting in hydrocompacted nanosilica (cA-300) [41], with increased bulk density (ρ_b_) toward 0.25 g/cm^3^, since the initial A-300 has ρ_b_ ≈ 0.05 g/cm^3^.

Dry PMS (dPMS) and dry cA-300 (1:1 w/w) powders were mixed in a porcelain mortar for 5 min without strong mechanical loading (sample 1, Bl). Distilled water (*h* = 0.1 g per gram of solids) was then added and the blend was stirred without (5 min, sample 2, Bh1l) or with (10 min, sample 3, Bh1s) strong mechanical loading. Water (*h* = 0.2 g/g) was added to S1, and the sample was stirred without strong mechanical loading (sample 4, Bh2l). Bh2l was additionally stirred with strong mechanical loading for 10 min (sample 5, Bh2s). Sample 6 (Adl) corresponds to dried and stirred cA-300 (Adl). Sample 7 (Pdl) corresponds to stirred dPMS. Sample 8 (Phdl) is hPMS-dried for a week (in sample labels: A is cA-300, B is the blend of dPMS and cA-300, l is low mechanical loading (~1 kg/cm^2^) upon simple mixing of samples, s is strong mechanical loading (~20 kg/cm^2^) upon mixing, d is dried samples, and h is hydrated samples at *h* = 0.1 g/g (h1) or 0.2 g/g (h2)).

### 2.2. Transmission (TEM) and Scanning (SEM) Electron Microscopy 

TEM (TECNAI G2 F30 microscope (FEI–Philips, Amsterdam, The Netherlands), operating voltage 300 kV) was used to analyze the particulate morphology of samples (Figure 1 and Appendix A). The powder samples were added to acetone (chromatographic grade) and sonicated. A suspension drop was then deposited onto a copper grid covered by a thin carbon film. After acetone evaporation, the dry sample remaining on the film was studied. 

SEM (FE–SEM, Hitachi S–4700, Tokyo, Japan, operating voltage of 15 kV, and magnification of ×5000–100000) (Figure 2, Appendix A) was used to analyze the morphological features of the dried powder samples. 

### 2.3. Textural Characteristics

To analyze the textural characteristics of individual and mixed samples degassed at 110 °C for 12 h (Table 1), low-temperature (77.4 K) nitrogen adsorption–desorption isotherms (Appendix A) were recorded using a Micromeritics ASAP 2460 adsorption analyzer (Micromeritics, Norcross, GA, USA). The specific surface area (Table 1, *S*_BET_) was calculated according to the standard BET method at 0.05 < *p*/*p*_0_ < 0.3 (using Micromeritics software), where *p* and *p*_0_ denote the equilibrium and saturation pressure of nitrogen at 77.4 K, respectively [42]. The total pore volume (Table 1, *V*_p_) was estimated from the nitrogen adsorption at *p*/*p*_0_ ≈ 0.98-0.99 [43]. The nitrogen desorption data were used to compute the pore-size distributions (PSD) (differential *f*_V_(*R*) ~ d*V*_p_/d*R* and *f*_S_(*R*) ~ d*S*/d*R*) using a self-consistent regularization (SCR) procedure under non-negativity condition (*f*_V_(*R*) ≥ 0 at any pore radius *R*) at a fixed regularization parameter α = 0.01 (Figure 3) [44]. A complex pore model including slit-shaped (S) and cylindrical (C) pores and voids (V) between spherical particles packed in random aggregates (SCV/SCR method) was applied [44]. The differential PSD with respect to the pore volume *f*_V_(*R*) ~ d*V*/d*R*, ∫*f*_V_(*R*)d*R* ~ *V*_p_ were recalculated to incremental PSD (IPSD) at Φ_V_(*R_i_*) = (*f*_V_(*R_i_*_+1_) + *f*_V_(*R_i_*))(*R*_i+1_ − *R*_i_)/2 at ∑ Φ_V_(*R_i_*) = *V*_p_) for a better view of the PSD at large *R* values. The *f*_V_(*R*) and *f*_S_(*R*) functions were used to calculate contributions of nanopores (*V*_nano_ and *S*_nano_ at 0.35 nm < *R* < 1 nm), mesopores (*V*_meso_ and *S*_meso_ at 1 nm < *R* < 25 nm), and macropores (*V*_macro_ and *S*_macro_ at 25 nm < *R* < 100 nm) [44]. The average values of the pore radii (Table 1) were determined with respect to the pore volume (*X* = *V*) and specific surface area (*X* = *S*) as the corresponding moments of the distribution functions:
(1)$$<R_{X}>=\int_{R_{\min}}^{R_{\max}}Rf_{X}(R)dR/\int_{R_{\min}}^{R_{\max}}f_{X}(R)dR$$

Additionally, the nonlocal density functional theory (NLDFT, Quantachrome software (Quantachrome Instruments, Boynton Beach, FL, USA), with a model of cylindrical pores in silica [45]) method was used to calculate the differential PSD (Appendix A). 

The main error in the evaluation of the textural characteristics such as the *S*_BET_ value based on the nitrogen adsorption isotherms is due to variation in the area (σ) occupied by nitrogen molecules. It is assumed that σ = 0.162 nm^2^. This value is correct for graphite, but in the case of silica, it is smaller [46] due nonparallel location of N_2_ molecules with respect to the surface plane. Therefore, overestimation of the *S*_BET_ value could be 16% for silica. Despite this fact, in firm software σ = 0.162 nm^2^, and for simplicity, one can use this value because the effective σ value for various materials is different and unknown. 

### 2.4. ^1^H MAS and ^29^Si CP/MAS NMR Spectroscopy

Solid-state ^1^H MAS NMR spectra (Figure 4a) were recorded using an Agilent DD2 600 MHz NMR spectrometer (Agilent, Santa Clara, CA, USA, magnetic field strength 14.157 T). A powder sample was placed in a pencil-type zirconia rotor of 4.0 mm o.d. The spectra were recorded at a spinning speed of 8 kHz with a recycle delay of 5 s. The adamantane was used as the reference of the ^1^H chemical shift of proton resonance (δ_H_). 

Solid-state ^29^Si CP/MAS NMR spectra (Figure 4b) were recorded using the same NMR spectrometer at a resonance frequency of 199.13 MHz for ^29^Si using the cross-polarization (CP), magic-angle spinning (MAS), and high-power ^1^H decoupling. The spectra were recorded at a spinning speed of 8 kHz (4 µ*s* 90° pulses), 2 ms CP pulse, and a recycle delay of 3 s. The Si signal of tetramethylsilane (TMS) at 0 ppm was used as the reference of the ^29^Si chemical shift (δ(^29^Si)) [2,47].

### 2.5. Infrared Spectroscopy

The infrared (IR) spectra (Figure 5 and Appendix A) were recorded in the range of 1800 to 300 cm^−1^ using a Specord M80 (Carl Zeiss AG, Oberkochen, Germany, 4 cm^−1^ step, integration time of 3 s, the transmission spectra were converted into absorbance ones). Samples were carefully ground (5 min) with KBr (Sigma-Aldrich, Saint Louis, MO, USA for spectroscopy) as 1:400 and pressed into thin pellets. 

## 3. Results and Discussion

Simple stirring (without strong mechanical loading) of the dry PMS and dry cA-300 powders (Bl) results in good distribution of small nanosilica aggregates in the composite (Figure 1a, Figure 2a, Appendix A). 

This composite is characterized by strong interactions between the PMS and silica nanoparticles because significant nonadditivity is observed for the textural characteristics of Bl in comparison to those of cA-300 (sample Adl) and dPMS (sample Pdl) (Table 1, Figure 3, Appendix A): *S*_BET_ = 186 m^2^/g instead of the expected (*S*_BET,S6_ + *S*_BET,S7_)/2 = 365 m^2^/g and *V*_p_ = 0.788 cm^3^/g instead of the expected 1.556 cm^3^/g. This result could be explained by the embedding of small aggregates of silica nanoparticles into soft PMS (as light structures observed in larger PMS structures, Figure 1a and Appendix A), and the grafting of PMS onto solid silica particles (similar to butter grafting onto macroporous bread resulting in a flat nonporous surface). If the silica aggregates could be remained at a surface of PMS particles that the strong decrease in the textural characteristics could be absent. The treatment changes the PSD in the total pore size range (Table 1, *S*_nano_, *V*_nano_, *S*_meso_, *V*_meso_, *S*_macro_, and *V*_macro_); and the contributions of pores of different shapes change too (Table 1, *c*_slit_, *c*_cyl_, and *c*_void_) in comparison to those of dry components (Adl and Pdl). A certain compaction of the blend is observed as a diminution of the <*R*_v_> value (Table 1). 

The nitrogen adsorption isotherms (Appendix A) correspond to type II according to the IUPAC classification [48]. The desorption hysteresis shape provides clear evidence for formation of constricted, textural mesopores in sample 6 (various voids between particles). However, this is less visible for other samples having a simple shape of the hysteresis loops. The fact that the hysteresis loop for sample 7 (dPMS) does not close, probably indicates irreversible swelling of this material during nitrogen adsorption step, as well the presence of long pores with a narrow throat formed upon drying of the PMS hydrogel that results in additional cross-linking of the polymers due to condensation of residual silanols.

The addition of a small amount of water (*h* = 0.1 g/g) to the dry blend and stirring without strong mechanical loading results in slightly better homogenization (better embedding of A-300 aggregates into PMS and better PMS grafting onto silica) of the blend (Appendix A), because water can play the role of a lubricant thus reducing the interactions of silica–silica and PMS–PMS. However, silica aggregates are visible in Bh1l as in Bl (Figure 1 and Appendix A).

There are certain changes in the textural characteristics of the Bh1l blend compared to Bl (Table 1, Figure 3, Appendix A), since both nanoporosity and mesoporosity decrease, but macroporosity increases due to reorganization of the secondary particles. 

An increase in mechanical loading (Bh1s) leads to significant homogenization of the blend (Appendix A), since silica aggregates (light structures) are not clearly visible as in Bl and Bh1l (Figure 1 and Appendix A). This can be explained by decomposition of the aggregates upon stirring and stronger embedding of individual silica nanoparticles into the PMS matrix. There is a certain decrease in the nanoporosity (Table 1, *S*_nano_ and *V*_nano_) as well an increase in the macroporosity (*S*_macro_, *V*_macro_, and <*R*_v_>). The contributions of pores of different shapes slightly change (Table 1, *c*_slit_, *c*_cyl_, and *c*_void_).

An increase in the amount of added water (*h* = 0.2 g/g) without strong mechanical loading (Bh2l) results in decreased porosity (water can fill the narrow pores and merge the particles together upon drying and degassing) (Table 1, Figure 3, Appendix A). However, the morphological changes are small (see Appendix A) in comparison to Bh1l. Quantitative analysis of SEM images of samples Bl and Bh2s as representatives (Appendix A) shows that the mechanical treatment of a wetted blend results in decomposition of aggregates. This is appropriate for compaction of the blend observed in the textural characteristics (Table 1). 

Enhanced mechanical loading (Bh2s) results in a strong diminution of the textural characteristics (Table 1), and the contributions of pores of different shapes change in comparison to Bh2l. Thus, upon stronger mechanical loading, the embedding-grafting effects increase in parallel to decomposition of silica aggregates, which are not visible similarly to Bh1s, but in contrast to Bl, Bh1l, and Bh2l. 

Treatment of the hydrogel hPMS (drying, degassing) can give (Phdl) very strong compaction of the system (Table 1, Appendix A). However, this compaction occurs in domains (aggregates) because the relative contributions of the macropores (*S*_macro_/*S*_BET_ and *V*_macro_/*V*_p_) are maximal for Phdl (this process is similar to the strong drying of wet soil with the appearance of checks).

The results of the treatment processes appear in the solid-state NMR (Figure 4, Appendix A) and IR (Figure 5) spectra. Treatment with the addition of water (resulting in better PMS grafting onto silica surface) reduces the intensity of the Q_2_–Q_4_ lines, well observed for Adl (cA-300) (Appendix A, Figure 4b).

The NMR spectra of water (δ_H_ = 4–5 ppm, Figure 4a) bound to PMS and the residual SiOH groups attached to =Si(CH_3_) (T_2_ lines in Figure 4b) depend on the type of samples and their treatment conditions. The amounts of water bound to dry PMS (samples Pdl and Phdl, Figure 4a) are relatively small because the amounts of residual SiOH groups (Figure 4b, T_2_ at −60 ppm), which are the main adsorption sites for bound water, are relatively small (Appendix A). The ^1^H MAS NMR spectrum of cA-300 (Adl) demonstrates three lines at δ_H_ ≈ 7 ppm (strongly disturbed silanols), δ_H_ ≈ 4–5 ppm (water and disturbed silanols), and 0–3 ppm (free silanols). Interactions of PMS with a silica surface lead to diminution of relative content of strongly disturbed silanols (e.g., for Bh2s, only one signal is observed at 4–5 ppm). For sample Bh2s, the lines of the CH_3_ groups (Figure 4a) and Q_2_–Q_4_ (Figure 4b) lines are not observed. 

The spectral changes and differences for the samples studied are observed in the IR spectra (Figure 5 and Appendix A). For example, samples of dry PMS (Pdl and Phdl) are characterized by practically identical bands of the symmetrical Si–O stretching vibrations at 800–700 cm^−1^ (Figure 5a). However, the asymmetrical Si–O stretching vibrations are at 1200–1000 cm^−1^ (especially at 1130–1115 cm^−1^) due to the difference in the cross-linking degree (compaction), the relative numbers of ≡SiO)_3_SiCH_3_ (T_3_, Figure 4b, Appendix A) and (≡SiO)_2_Si(OH)CH_3_ (T_2_), and the compaction type on drying (stronger check effects on drying of hPMS, Table 1). Water is stronger bound and in a larger amount in the blend than in dPMS alone (Figure 5c). The stirring effects more weakly appear in the IR spectra (Figure 5b) than in the textural characteristics or NMR spectra, because the IR spectra are transmission spectra of the total samples. The PMS-onto-silica grafting/embedding effect is therefore difficult to observe in the IR spectra. 

Theoretical modeling (see Appendix A) shows that PMS particles may be hydrated due to the presence of residual silanol groups appearing in the ^29^Si CP/MAS NMR spectra (Figure 4b) as a line at −60 ppm. This effect explains the good distribution of hydrated silica nanoparticles in the hydrated blends after significant mechanical loading (Figure 1 and Appendix A). Note that the use of relatively small amounts of water (*h* = 0.1 or 0.2 g/g) can prevent the formation of separate phases of PMS and A-300 upon mechanical treatment of the blends. 

## 4. Conclusions

The effects of adding amounts of water (0–0.5 g of water per gram of dry solids) to the polymethylsiloxane/fumed silica blends (1:1 w/w) undergoing different mechanical treatment (stirring with weak (~1–2 kg/cm^2^) and strong (~20 kg/cm^2^) loading) were analyzed using TEM and SEM images, nitrogen adsorption–desorption isotherms (pore size distribution and other textural characteristics), solid-state NMR (^1^H MAS and ^29^Si CP/MAS) spectra, and infrared spectra in the range of 1700 to 300 cm^−1^ (related to the Si–O stretching vibrations). Both dry and wetted PMS (with 3D cross-linking in the particles containing residual SiOH and CH_3_ group attached to each Si atoms) is a soft material, which can be grafted onto a silica surface, and silica aggregates can be embedded into the PMS matrix even upon weak mechanical loading. The main evidence of this effect is strongly nonadditive changes in the textural characteristics, which become two to three times smaller than additive those expected. All the PMS/nanosilica blends, demonstrating a good distribution of nanosilica nanoparticles and their small aggregates in the polymer matrix, are rather meso/microporous, with the main PSD peaks at *R* > 10 nm and average <*R*_V_> values of 18-25 nm. Contributions of nanopores (*R* < 1 nm), mesopores (1 nm < *R* < 25 nm), and macropores (25 nm < *R* < 100 nm) depend strongly on the type of the materials (A-300, cA-300, dPMS, dried hPMS, and dPMS/cA-300 blends dried or wetted) and the type of treatment and amounts of added water. Thus, the developed technique allows one to significantly change the morphological and textural characteristics of fumed silica (hydrocompaction), PMS (drying–wetting–drying), and PMS/A-300 blends (wetting–drying under controlled mechanical loading), something that is of importance from a practical point of view. 

## Figures and Tables

**Figure 1 materials-12-02409-f001:**
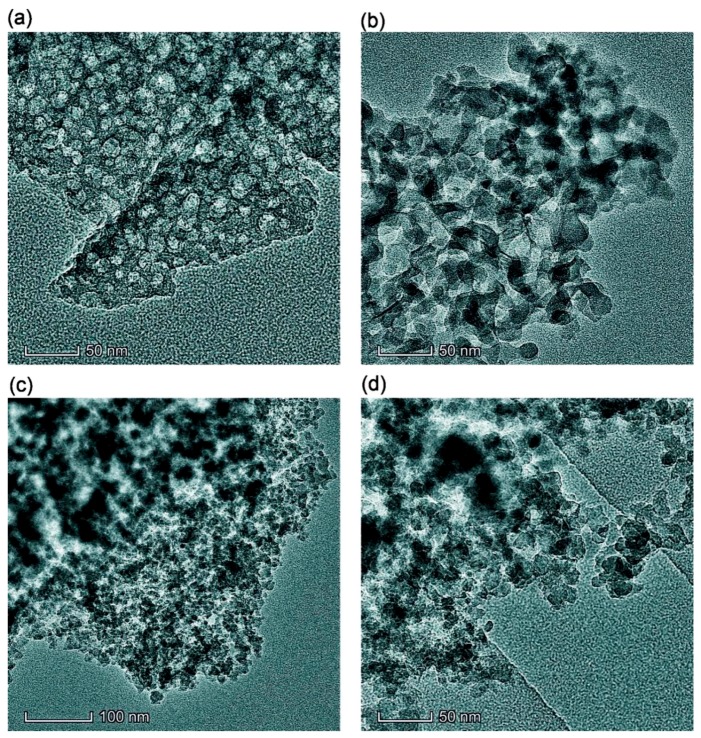
TEM images of samples (**a**) Bl, (**b**) Bh2s, (**c**) Adl, and (**d**) Pdl (scale bar 50 nm (a, b, d) and 100 nm (c)).

**Figure 2 materials-12-02409-f002:**
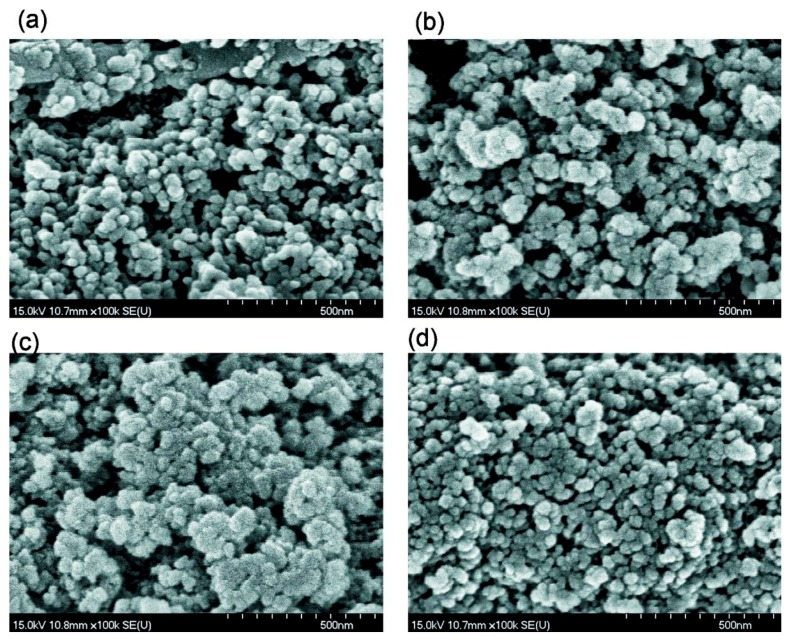
SEM images of (**a**) Bl, (**b**) Bh2s, (**c**) Adl, and (**d**) Pdl (scale bar 500 nm).

**Figure 3 materials-12-02409-f003:**
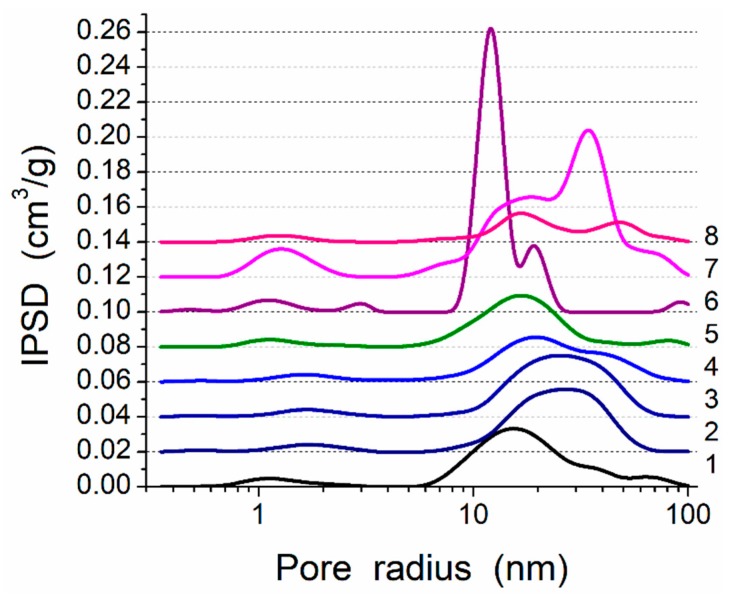
Incremental pore-size distributions (PSD) (SCV/SCR method) for PMS, A-300 and their blends (curve numbers correspond to sample numbers in Table 1).

**Figure 4 materials-12-02409-f004:**
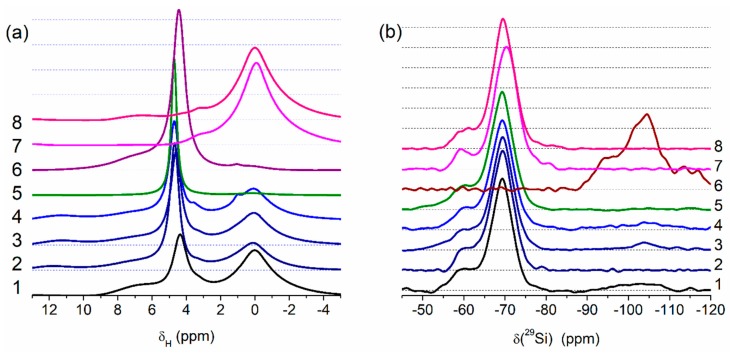
Solid state NMR spectra of (**a**) ^1^H (MAS) and (**b**) ^29^Si (CP/MAS) for PMS, A-300 and their blends (curve numbers correspond to sample numbers) (Q_n_ corresponds to Si(OH)_4−n_(OSi≡)_n_ at *n* = 2 (−91~−93 ppm), 3 (−99~−102 ppm), and 4 (−109~−111 ppm); T_3_ (−70 ppm) corresponds to (≡SiO)_3_SiCH_3_) and T_2_ (−60 ppm) corresponds to (≡SiO)_2_Si(OH)CH_3_), see Appendix A).

**Figure 5 materials-12-02409-f005:**
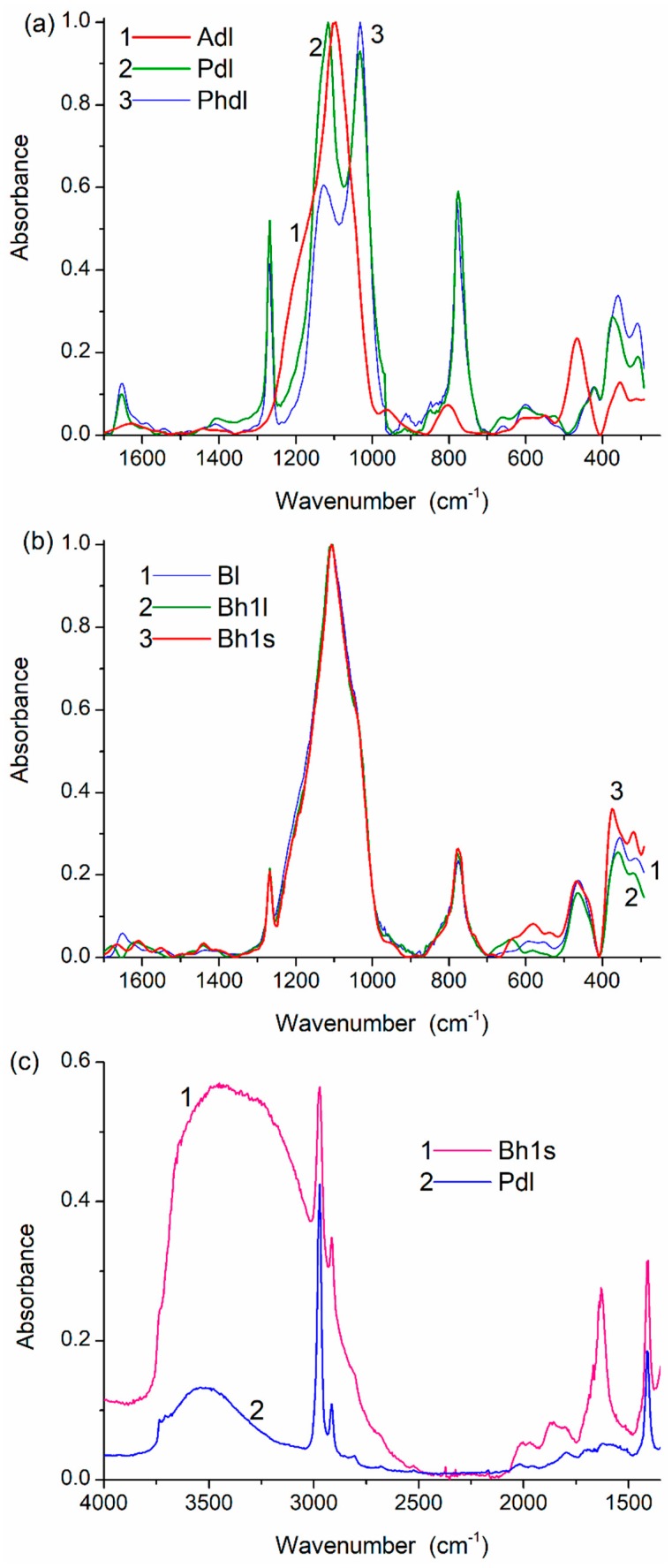
IR spectra of (**a**,**c**) PMS, A-300 and (**b**,**c**) their blends in the range of (**a**,**b**) 1700–300 cm^−1^ and (**c**) 4000–1300 cm^−1^.

**Table 1 materials-12-02409-t001:** Textural characteristics of PMS, A-300, and their blends (SCV/SCR method).

No	Sample Composition	Sample Label	*S*_BET_ (m^2^/g)	*S*_DFT,cyl_ (m^2^/g)	*S*_nano_ (m^2^/g)	*S*_meso_ (m^2^/g)	*S*_macro_ (m^2^/g)	*V*_p_ (cm^3^/g)	*V*_nano_ (cm^3^/g)	*V*_meso_ (cm^3^/g)	*V*_macro_ (cm^3^/g)	<*R*_V_> (nm)	<*R*_S_> (nm)	*c* _slit_	*c* _cyl_	*c* _void_
1	dPMS/cA-300	Bl	186	164	52	124	10	0.788	0.026	0.594	0.168	19.65	7.57	0.699	0.221	0.080
2	dPMS/cA-300	Bh1l	166	152	38	107	22	0.827	0.012	0.463	0.352	24.17	9.69	0.162	0.798	0.041
3	dPMS/cA-300	Bh1s	165	151	34	108	23	0.848	0.011	0.461	0.376	25.13	10.27	0.171	0.774	0.055
4	dPMS/cA-300	Bh2l	159	150	31	113	14	0.691	0.011	0.433	0.247	23.51	8.44	0.200	0.740	0.060
5	dPMS/cA-300	Bh2s	134	127	32	97	5	0.613	0.018	0.506	0.089	18.37	7.60	0.775	0.136	0.089
6	cA-300	Adl	278	278	77	201	1	1.304	0.038	1.243	0.024	13.39	7.19	0.794	0.190	0.016
7	dPMS	Pdl	453	459	83	319	52	1.808	0.049	0.916	0.843	25.26	7.86	0.396	0.302	0.302
8	PMS initial	Phdl	98	99	19	71	8	0.390	0.011	0.234	0.145	25.41	7.62	0.437	0.249	0.315
9	A-300 initial	A-300	294	289	44	229	16	0.850	0.023	0.567	0.259	20.41	6.14	0.379	0.280	0.341

Note: The values of *V*_nano_ and *S*_nano_, *V*_meso_ and *S*_meso_, and *V*_macro_ and *S*_macro_ were calculated by integration of the *f*_V_(*R*) and *f*_S_(*R*) functions at 0.35 nm < *R* < 1 nm, 1 nm < *R* < 25 nm, and 25 nm < *R* < 100 nm, respectively. The values of <*R*_V_> and <*R*_S_> as the average pore radii were calculated as a ratio of the first moment of *f*_V_(*R*) or *f*_S_(*R*) to the zero moment (integration over the 0.35 to 100 nm range) <*R*> = ∫*f*(*R*)*RdR*/∫*f*(*R*)*dR*. *D*_f_ is the fractal dimension (Frenkel–Hill–Halsey method); *c*_slit_, *c*_cyl_, and *c*_void_ are the weight constants calculated using the SCV/SCR method.

## Data Availability

The datasets supporting the conclusions of this work are included within the article. Any raw data generated and/or analyzed in the present study are available from corresponding author on request.

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
