# Peer review of "Nanostructured Polymethylsiloxane/Fumed Silica Blends"

_materials, 2019, doi:10.3390/ma12152409_

Round 1

Reviewer 1 Report

I suggest that the authors attach a table describing the acronyms used for the different samples and even if it is possible to change the names from S1 to S8 by others related to their composition and/or treatment. This symbology complicates the reading of the article

Author Response

We thank the reviewer for his careful scrutiny of the manuscript. We are very appreciative of the time that he has taken to consider this submission.

Reviewer 1

Comments and Suggestions for Authors:

I suggest that the authors attach a table describing the acronyms used for the different samples and even if it is possible to change the names from S1 to S8 by others related to their composition and/or treatment. This symbology complicates the reading of the article.

Authors: It was done. Sample labels were changed and described (and collected in Table S1).

Reviewer 2 Report

Protsak et al blended polymethylsilosane with fumed silica and characterized with different techniques. The manuscript lacks the discussion section. The authors should pay attention on the followings during revision;

Why is it necessary to blend polymethylsilosane with fumed silica?

The experimental part should be elaborated. What is strong mechanical loading? It should be written in scientific language.

Authors claimed that after blending; nanoporosity and mesoporosity decrease but macroporosity increases. What is the reason behind it?

What is the practical application of the materials?

Author Response

We thank the reviewer for his careful scrutiny of the manuscript. We are very appreciative of the time that he has taken to consider this submission.

Reviewer 2

Comments and Suggestions for Authors:

Protsak et al blended polymethylsilosane with fumed silica and characterized with different techniques. The manuscript lacks the discussion section. The authors should pay attention on the followings during revision;

·         Why is it necessary to blend polymethylsilosane with fumed silica?

Authors: Additional explanation was added into Introduction. As a whole, the blends were used for better control of the textural characteristics of medicinal sorbents, which are also used as drug carriers.

·         The experimental part should be elaborated. What is strong mechanical loading? It should be written in scientific language.

Authors: Different mechanical treatments correspond to stirring of the samples with low (~1-2 kg/cm2) and strong (~20 kg/cm2) loading.

·         Authors claimed that after blending; nanoporosity and mesoporosity decrease but macroporosity increases. What is the reason behind it?

Authors: This aspect is of importance for medicinal sorbents and drug carriers because variations in the pore size distributions affect both adsorption (e.g. toxins, etc.) and drug release.

·         What is the practical application of the materials?

Authors: These materials (cA-300, Enterosgel, PMS) are used as (i) medicinal sorbents; (ii) drug carriers; and (iii) food additives. This study was directed toward improvement of some important properties of these materials using them in blends.

Reviewer 3 Report

This manuscript is a study on the influence of mixing parameters on the structure of polymethylsiloxane/silica blends. The soft polymer coats the surface of the silica particles and fills its pores partially, resulting in a decrease in pore volume.  The synthetic approach (mixing both components with various amounts of water with the help of mortar) is rather simple, with electron microscopy and nitrogen adsorption measurements applied as the main analytical methods, which support the rather obvious conclusion of the study. Additionally, MAS NMR, IR and quantum chemical calculations were used for gaining insight in changes on a molecular level.

While the manuscript is well structured and most results are presented in a logical fashion, there are major revisions/improvements necessary for making the suitable for publication.

1)      The authors claim practical importance of their findings, but it is never clearly stated for which applications these composites and, especially, pore sizes are relevant. The authors are only making very general statements, such as,    “widely used in industry and medicine, as well as various branches of science” (line 45-46), “importance from a practical point of view” (lines 31, 272). The manuscript would benefit from examples of these applications and how the changes in the parameters studied are relevant.

2)       Since the mechanical load is one of the key parameters varied, how was the mechanical load measured and guaranteed that this parameter was consistent between similar samples? This is especially relevant, since some of the changes observed are rather small (see 3) and 4)).

3)      I can’t see much difference in the TEM/SEM pictures. Since both starting materials look rather similar, changes in morphology are hard to observe. In addition, it is possible that the differences are due to slightly different focus (e.g. the out of focus area in the top left of Fig. 1d, looks similar to Fig. 1b). The study would benefit from more careful investigation of the morphology, by combining electron microscopy with elemental mapping or using scattering techniques to investigate how well the particles are dispersed. Furthermore, using always the same magnification in a series would make it much easier to compare the pictures.

4)      The changes in pore area and volume are rather small (e.g. Sa2, Sa3, Sa4, Sa5). How significant are these differences, how large is the error of the analytical methods used and how much difference do different batches of materials differ? With only minor changes observed with other analytical methods, it might be possible that the mixing parameters have no influence at all on the properties of the composite.

5)      It is quite hard to figure out which sample number corresponds to which sample. The manuscript would benefit from a table summarizing the different mixing parameters.

6)      Spelling of polymethylsiloxane wrong in line 2.

7)      What type of stability is meant in line 59, chemical stability of mechanical stiffness?

Author Response

We thank the reviewer for his careful scrutiny of the manuscript. We are very appreciative of the time that he has taken to consider this submission.

Reviewer 3

Comments and Suggestions for Authors:

This manuscript is a study on the influence of mixing parameters on the structure of polymethylsiloxane/silica blends. The soft polymer coats the surface of the silica particles and fills its pores partially, resulting in a decrease in pore volume.  The synthetic approach (mixing both components with various amounts of water with the help of mortar) is rather simple, with electron microscopy and nitrogen adsorption measurements applied as the main analytical methods, which support the rather obvious conclusion of the study. Additionally, MAS NMR, IR and quantum chemical calculations were used for gaining insight in changes on a molecular level.

While the manuscript is well structured and most results are presented in a logical fashion, there are major revisions/improvements necessary for making the suitable for publication.

1)      The authors claim practical importance of their findings, but it is never clearly stated for which applications these composites and, especially, pore sizes are relevant. The authors are only making very general statements, such as,    “widely used in industry and medicine, as well as various branches of science” (line 45-46), “importance from a practical point of view” (lines 31, 272). The manuscript would benefit from examples of these applications and how the changes in the parameters studied are relevant.

Authors: Additional explanation was added into Introduction. As a whole, the blends were used for better control of the textural characteristics of medicinal sorbents, which are also used as drug carriers. The textural characteristics are of importance for medicinal sorbents and drug carriers because variations in the pore size distributions, specific surface area, pore volume, etc. affect both adsorption (e.g. toxins, etc.) and drug release.

2)       Since the mechanical load is one of the key parameters varied, how was the mechanical load measured and guaranteed that this parameter was consistent between similar samples? This is especially relevant, since some of the changes observed are rather small (see 3) and 4)).

Authors: Different mechanical treatments correspond to stirring of the samples with low (~1-2 kg/cm2) and strong (~20 kg/cm2) loading (these values were estimated from the geometry of the used a porcelain mortar and applied forces). Note that the applied forces are too small to destroy nanoparticles of A-300 (this effect was observed upon cryogelation of mixed fumed oxides (core-shell nanoparticles of 50-200 nm in size) in cryoboms at ~1000 atm, but even at this pressure simple nanoparticles of A-300 were remained without decomposition). This mechanical treatment was used to reorganize aggregates of nanoparticles and agglomerates of aggregates, as well to prepare relatively uniform blends with A-300 and PMS at stronger loading (~20 kg/cm2).

3)      I can’t see much difference in the TEM/SEM pictures. Since both starting materials look rather similar, changes in morphology are hard to observe. In addition, it is possible that the differences are due to slightly different focus (e.g. the out of focus area in the top left of Fig. 1d, looks similar to Fig. 1b). The study would benefit from more careful investigation of the morphology, by combining electron microscopy with elemental mapping or using scattering techniques to investigate how well the particles are dispersed. Furthermore, using always the same magnification in a series would make it much easier to compare the pictures.

Authors: Changes in the organization of aggregates and agglomerates are better controlled using the nitrogen adsorption method because it gives a picture on total weight of a sample, but TEM and SEM give a picture of a small portion of the sample. Quantitative analysis of SEM images (new Fig. S4) shows that the particle (aggregates) size distributions depend on hydration and mechanical loading.

4)      The changes in pore area and volume are rather small (e.g. Sa2, Sa3, Sa4, Sa5). How significant are these differences, how large is the error of the analytical methods used and how much difference do different batches of materials differ? With only minor changes observed with other analytical methods, it might be possible that the mixing parameters have no influence at all on the properties of the composite.

Authors: The differences observed in the data depend on the tested weight (or a part) of the materials. Microscopic images give a picture only for a small part of the studied sample in contrast to the adsorption method. Additionally, the PSD calculated using the nitrogen adsorption data give only a portion of macropores. For example, the empty volume calculated from the bulk density is ~25 and ~5 cm3/g for A-300 and cA-300, respectively, but the Vp values are much smaller because nitrogen poorly fills large macropores (voids) in the powders having low bulk density.

5)      It is quite hard to figure out which sample number corresponds to which sample. The manuscript would benefit from a table summarizing the different mixing parameters.

Authors: It was done.

6)      Spelling of polymethylsiloxane wrong in line 2.

Authors: It was corrected.

7)      What type of stability is meant in line 59, chemical stability of mechanical stiffness?

Authors: This is both chemical stability and mechanical stiffness (additional information was added into the MS)

Round 2

Reviewer 2 Report

Accept in the present form. Congrats.

Author Response

We have done minor changes/spelling in the text according to the reviewer's concern. 

We thanks to the reviewer very much for the time he has taken to review our manuscript.

Reviewer 3 Report

The authors tremendously improved the readability and presentation quality with the revised version. Most of the points criticized in the first version were addressed appropriately.  

The only point not addressed sufficiently is the errors of the nitrogen adsorption measurements. I still think that the magnitude of error for a method should be mentioned when discussing –what seems to me- rather small differences. This is especially important since a lot of readers might not be familiar with a technique like nitrogen adsorption.  

Author Response

We agree with the reviewer that it is indeed important to mention the errors of the BET measurements and we have done it on the page 4 in the section of BET measurements in the revised MS. We also have provided the reference according to this context. We thank the reviewer for focusing us on the important points in the manuscript which were needed to be revised.